# One Month into the Reinforcement of Social Distancing due to the COVID-19 Outbreak: Subjective Health, Health Behaviors, and Loneliness among People with Chronic Medical Conditions

**DOI:** 10.3390/ijerph17155403

**Published:** 2020-07-27

**Authors:** Roni Elran-Barak, Maya Mozeikov

**Affiliations:** School of Public Health, Faculty of Social Welfare and Health Sciences, University of Haifa, Haifa 3498838, Israel; smayal@hotmail.com

**Keywords:** COVID-19, self-rated health, health behaviors, loneliness, chronic illness

## Abstract

We sought to examine how the near-lockdown measures, announced by the Israeli government in an effort to contain the COVID-19 outbreak, impacted the self-rated health (SRH), health behaviors, and loneliness of people with chronic illnesses. An online cross-sectional survey was carried out about one month (20–22 April 2020) after the Israeli government reinforced the severe social distancing regulations, among a convenience sample of 315 participants (60% women) with chronic conditions (27% metabolic, 17% cardiovascular, 21% cancer/autoimmune, 18% orthopedic/pain, 12% mental-health). Results suggested that about half of the participants reported a decline in physical or mental SRH, and as many as two-thirds reported feeling lonely. A significant deterioration in health behaviors was reported, including a decrease in vegetable consumption (*p* = 0.008) and physical activity (*p* < 0.001), an increase in time spent on social media (*p* < 0.001), and a perception among about half of the participants that they were eating more than before. Ordinal regression suggested that a decline in general SRH was linked with female gender (*p* = 0.016), lack of higher education (*p* = 0.015), crowded housing conditions (*p* = 0.001), longer illness duration (*p* = 0.010), and loneliness (*p* = 0.008). Findings highlight the important role of loneliness in SRH during the COVID-19 lockdown period. Future studies are warranted to clarify the long-term effects of social-distancing and loneliness on people with chronic illnesses.

## 1. Introduction

The World Health Organization (WHO) declared the coronavirus 2019 (COVID-19) outbreak an international public health emergency on January 30, 2020 and a pandemic on March 11, 2020 [1]. In an effort to contain the COVID-19 outbreak, the Israeli government announced a number of new restrictions aimed at reinforcing social distancing [2]. On March 12, Israel announced that all universities and schools would close. On March 16, all non-critical government and local authority workers were placed on paid leave, and private sector firms were required to reduce the staff presence in the workplace [3]. On March 19, Prime Minister Benjamin Netanyahu declared a national state of emergency [4], saying that existing restrictions would henceforth be legally enforceable, and violators would be fined. Israelis were not allowed to leave their homes unless absolutely necessary, putting a near-lockdown into effect. Essential services—including grocery stores, pharmacies, and banks—remained open, but people were prohibited from venturing more than 100 m from their homes, apart from under certain circumstances (e.g., stocking up on food and medicine). Non-essential stores were required to close, and parks were to remain shut. People were required not to participate in any social gatherings and to limit face-to-face interactions with individuals outside the immediate household [5].

Self-rated health (SRH)—also known as subjective health, self-assessed health, or perceived health—is a simple and straightforward construct that has a strong predictive capacity regarding various health-related outcomes, including mortality [6,7,8,9]. Numerous studies suggest that, among people with chronic illnesses, SRH encompasses many of the objective aspects of health including clinical, cognitive, and functional aspects [10,11,12,13]. For example, studies among individuals with Type 2 diabetes suggest that SRH is highly associated with clinical status (e.g., glycemic control, BMI, and blood pressure) [14,15], and provides additional information regarding mortality risk, independent of demographic, socioeconomic, and medical risk factors. In addition, studies about changes in SRH suggest that a decline (or improvement) in SRH can independently predict better (or worse) long-term prognoses among people with chronic illnesses [16]. Therefore, and based on the extant literature highlighting the importance of SRH and changes in SRH in the assessment and understanding of the medical status of people with chronic illnesses, we aimed to investigate whether people with chronic illnesses experienced changes in SRH after one month of the social distancing reinforcement due to the coronavirus outbreak. Changes in health behaviors (e.g., physical activity, food consumption) were also assessed, as these behaviors may have been affected by the lockdown situation [3]. It is important to assess changes in health behaviors given that these behaviors can have detrimental effects on health [17], especially among people with chronic medical conditions [18]. A recent study conducted in France and Switzerland among the general population assessed whether changes in physical activity and sedentary behaviors during the COVID-19 lockdown were associated with changes in mental and physical health [19]. Results of this study showed that the lockdown in these countries resulted in an increase in sedentary behaviors, which was associated with a decrease in physical health, mental health, and subjective vitality [19]. Although no research has yet been conducted regarding the effect of the COVID-19 crisis on health behaviors in people with chronic illnesses, it may be presumed that the disruption in usual activity would be even more harmful in this population.

Loneliness is an important determinant of health [20], and studies suggest that loneliness and social isolation are major risk factors for morbidity and mortality, risk factors comparable in importance to obesity, sedentary lifestyles, and possibly even smoking [21]. For example, a longitudinal study conducted among a U.S. nationally representative sample reported that loneliness was associated with an increased mortality risk [22]. Specifically, among people with chronic illnesses, severity of illness can predict enhanced feelings of loneliness, and loneliness, in turn, can aggravate the medical condition [23]. Nevertheless, despite the understanding that lockdown measures can increase loneliness, specifically among vulnerable populations [24], studies examining the impact of loneliness on the health of people with chronic medical conditions during the COVID-19 outbreak are, as yet, scarce.

We also aimed to examine factors contributing to changes in SRH. The literature suggests that various factors, including sociodemographic and medical-related factors, can contribute to changes in SRH [25]. For example, there is some evidence that lower socioeconomic status, as reflected for instance by lower education level, or higher number of people living in the household, is often correlated with worse SRH [26]. Furthermore, females may be more vulnerable to psychological distress than are males [27], and this psychological distress may be linked with a stronger decline in mental SRH during stressful occasions (e.g., the COVID-19 outbreak). Indeed, Mazza et al. [28] found that female gender was associated with higher levels of depression, anxiety, and stress during the COVID-19 lockdown in Italy. In addition, studies among patients with chronic illnesses suggest that those with higher illness severity, as represented for instance in higher BMI, more medical visits, and longer illness duration, are more vulnerable to a decline in SRH [10,11,12]. Therefore, in the current study we sought to find those characteristics of people with chronic illnesses that might be linked with a stronger decline in SRH.

An online survey was conducted one month into the reinforcement of social distancing (between 20 April and 22 April 2020) among people with various chronic medical conditions in order to address the following research questions:Did people with chronic illnesses experience changes in their SRH, one month into the reinforcement of social distancing?Did people with chronic illnesses report changes in loneliness and health behaviors, one month into the reinforcement of social distancing?Did people with different chronic illnesses experience different changes in self-rated health, loneliness, and health behaviors one month into the reinforcement of social distancing?Were there specific characteristics, such as sociodemographic and medical-related factors, that were linked with a decline in SRH one month into the reinforcement of social distancing among people with chronic illnesses?

It is important to address these questions, as chronic medical conditions may increase the risk of having a fatal reaction to COVID-19 [29,30,31,32]. That is, these vulnerable individuals (who were advised to take extra social-distancing measures) are under additional stresses and likely more confused about how to handle their ongoing medical issues in the context of COVID-19 [1]. Relatedly, the fact that non-emergency medical services were completely shut down or reduced to an absolute minimum during the examined period must have been particularly challenging for people with chronic illnesses, who have no choice but to make use of these services on a regular basis. Hence, findings from the current study may help us understand how the reinforcement of social distancing might have influenced the psychological and physical health [33,34,35,36] of people with chronic illnesses, in order to inform policymakers about the effect of social distancing on the health of vulnerable populations.

## 2. Methods

### 2.1. Procedure 

Information was collected using an online, self-report survey, which was advertised through Camoni (Hebrew for “like me”, http://www.camoni.co.il/). Camoni is the first Israeli medical social network to have been established, and it includes several virtual health communities for people who share similar medical conditions, such as diabetes, cancer, pain, depression, or eating disorders. These are moderated communities—maintained by a governmental organization, the Gertner Institute for Epidemiology and Health Policy Research (http://www.gertnerinst.org.il/)—that are free of charge and accessible to everyone. Users can log into the communities to share in-the-moment feelings or to seek momentary support from moderators and other users. The communities do not present themselves or intend to be seen as alternatives to professional treatment, and the moderators of the communities encourage community users to receive professional treatment. The invitation to participate in the study was advertised by the maintainer of Camoni through a newsletter, thereby using a convenience-sampling method. 

On the first page of the online survey and prior to the start of the questionnaire, participants were asked to press “continue” if they were over the age of 18 and agreed to participate in the survey. Participants were also informed that they could choose not to participate in the study or to stop participating at any stage. In addition, they were told that the questionnaire was anonymous, and that no personal information would be collected (e.g., name or address). The contact details of the primary investigator (R.E.B.) were provided, as well as the name of the funding organization. Following consent, participants were provided with a link to a designated website which contained the anonymous survey. Participants who wished to participate in a lottery were invited to provide their contact information (e.g., email/telephone) at the end of the survey. A 50-shekel (Israeli currency) cash gift was the prize, equivalent to about 15 U.S. dollars. The study was approved by the university’s institutional review board IRB. Ethical approval number 042011.

### 2.2. Statistical Analyses

Data were analyzed using IBM (New York, NY, USA) SPSS-25. Chi^2^ and one-way ANOVA with LSD post-hoc tests were conducted to examine differences among the five medical condition categories. Paired T-tests were conducted to examine changes in health behaviors. Ordinal logistic regressions were fitted to examine which factors contributed to perceived changes in physical, mental, and general health occurring during the first month of the outbreak. The orthopedic/pain-related condition was selected as a reference group in the regression models, given that univariate analyses suggested that people with this condition reported the greatest decline in SRH. The independent variable “crowded living” was dichotomized in the regression models to not at all (74.5%) versus somewhat (25.5%), as its distribution was not normal.

### 2.3. Measures

Individual and Socioeconomic Variables: These variables consisted of gender (female, male), age group (18–25, 26–35, 36–45, 46–55, 56–65, 66–75, 76–85, >85 years), marital status (single, married or live with a partner, divorced, separated, widowed), education (12 years or less, vocational education, college/university degree), work status during and before the COVID-19 crisis (full-time, part-time, unemployed, on sick leave, retired), economic status (bad, very bad, pretty good, good, very good), religiosity (secular, traditional, religious), number of people in household (1, 2, 3, >3), and perception of crowded living (“To what extent do you agree with the following statement: I live in crowded housing conditions” with five response options from completely disagree [1] to completely agree [5]). Categories containing few responses were combined.

#### 2.3.1. Medical Condition

Participants self-selected their one main medical condition from a list of predefined conditions, based on the different communities within the Camoni platform, with an option to add an additional condition not mentioned in the list. In addition, participants reported the duration of their medical condition (less than one, one to three, three to five, more than five years), whether they generally received medical care for their medical condition (yes/no), and their Body Mass Index (BMI, Kg/m^2^). In addition, one item was included to examine frequency of medical appointments/visits: “How often did you meet with your medical team before the COVID-19 crisis (daily, weekly, monthly, bi-yearly, yearly, less than once a year)?

#### 2.3.2. Health Behaviors

Participants were asked to report the frequency of their health-related behaviors, during and before the COVID-19 crisis via the following questions—“On average, how many times a week do you participate in any exercise/sports activity for half an hour or longer?” This item was based on the 36-item short-form (SF-36) Medical Outcomes Study (MOS) [37]. “On average, how many times a day do you eat fresh fruit?” “On average, how many times a day do you eat fresh vegetables?”—with answers ranging from 0–10 times per day, based on the 2-item Serving Fruits and Vegetables Scale (2-Serving FVS) [38]. Participants were also asked to compare the amount of food they consumed during and before the COVID-19 crisis, with answers ranging from 1 (a lot more than before) to 5 (a lot less than before).

#### 2.3.3. Disease Management

Disease management was defined to participants as actions that patients need to take routinely in order to maintain their health, for example, going to medical appointments, adhering to nutrition recommendations, taking medications, and engaging in physical activity. Perceived disease management, during and before the COVID-19 crisis, was measured by a single item: “In my opinion, I manage my illness optimally,” rated on a 5-point Likert scale with answers ranging from 1 (strongly disagree) to 5 (strongly agree). This item was adopted from previously validated disease-specific self-efficacy measures [39] and from the Challenges to Illness Management Scale [40].

#### 2.3.4. Time Spent on Social Media

Total hours per day spent on social media, during and before the COVID-19 crisis, were measured by two single items: “How many hours per day, on average, do you spend on social media in general?” and “How many hours per day, on average, do you spend on online health communities to receive or provide information about the medical condition you are dealing with?” Participants were asked to answer this question in regard to the week before the outbreak and the previous week. Hours per day were measured on a seven-point scale: (0; <1; 1–2; 2–4; 4–6; 6–10; >10 hours per day). These items were adopted from the Technology Use Questionnaire [41,42].

#### 2.3.5. Self-Rated Health

Five items were included to examine SRH before and during the outbreak, revised from the SF-36 MOS [37] and adapted for the COVID-19 pandemic: (1) “In your opinion, as of today, your physical health condition is?” with answers ranging from 1 (very bad) to 5 (very good). (2) “In your opinion, as of today, your mental health condition is?” with answers ranging from 1 (very bad) to 5 (very good). (3) “Has your medical condition improved or worsened in the last month?” with answers ranging from 1 (greatly improved) to 5 (greatly worsened). (4) “How much do you feel that the COVID-19 crisis has affected your physical health?” with answers ranging from 1 (greatly improved) to 5 (greatly worsened). (5) “How much do you feel that the COVID-19 crisis has affected your mental health?” with answers ranging from 1 (greatly improved) to 5 (greatly worsened).

#### 2.3.6. Loneliness

The 3-item version of the Revised UCLA Loneliness Scale [43,44] was used to assess participants’ sense of loneliness, during and before the COVID-19 crisis. The Hebrew-translated version of the scale was used [45]. Items are as follows: “How often do you feel that you lack companionship?” “How often do you feel left out?” and “How often do you feel isolated from others?” with answers ranging from 1 (hardly ever) to 2 (some of the time) or 3 (often). The three items are summed to form a total score ranging from 3-9. Participants were asked to rate their current and past loneliness (i.e., before the pandemic) using these three items.

### 2.4. Participants

Participants comprised 315 individuals (60% were women). Inclusion criteria included subjects over the age of 18 with a chronic medical condition, who use the Camoni website for themselves and not for another person (e.g., for a family member). One quarter of the participants were aged 18–45 years, and the rest were older (14% were 46–55, 22% were 56–65, 34% were 66–75, and 11% were older than 76 years). The average BMI was 27.4 Kg/m^2^. More than half (57%) of the participants reported being unemployed before the COVID-19 crisis compared to as many as 70% one month into the crisis. Specifically, before the crisis, a quarter (25%) of participants worked full-time away from home, and 3% worked full-time at home, compared to 9% away from home and 7% at home after the crisis; 14% worked part-time away from home and 2% worked part-time at home before the crisis, compared to 7% away from home and 9% at home after the crisis; 3% were on sick leave; 10% were unemployed, and 41% were retired before the crisis. After the crisis, 17% were on unpaid leave and 2% were let go due to the crisis; about half (48%) reported not working before the crisis or currently. As many as three quarters (74.5%) of the participants reported that they completely disagreed with the statement “I live in crowded housing conditions” and the rest reported that they partly disagreed (10.7%), did not agree/disagree (6.9%), partly agreed (1.4%) or completely agreed (5.2%). Additional characteristics of the study sample are presented in Table 1.

Participants had one of six medical conditions: (1) Mental health conditions (*n* = 36, 12%): depression and anxiety (*n* = 25), eating disorders (*n* = 3), posttraumatic stress disorder (*n* = 2), attention deficit disorder (*n* = 4), schizoaffective disorder (*n* = 1), and personality disorder (*n* = 1). (2) Metabolic conditions (*n* = 83, 27%): obesity (*n* = 16), diabetes (*n* = 58), kidney disease (*n* = 5), digestive issues (*n* = 2), liver disease (*n* = 1), thyroid conditions (*n* = 1). (3) Cardiovascular conditions (*n* = 53, 17%): heart-related diseases (*n* = 33), blood pressure disorders (*n* = 17), stroke (*n* = 3). (4) Cancer and autoimmune conditions (*n* = 64, 21%): cancer (all types) (*n* = 18), Crohn’s disease and ulcerative colitis (*n* = 11), multiple sclerosis (*n* = 17), psoriasis (*n* = 5), Sjogren’s (*n* = 2), Lupus (*n* = 4), HIV (*n* = 1), rheumatic diseases (*n* = 5), endometriosis (*n* = 1), Guillain-Barré (*n* = 1). (5) Orthopedic/pain-related conditions (*n* = 54, 18%): orthopedic conditions (*n* = 21), osteoporosis (*n* = 12), osteoarthritis (*n* = 4), pain (*n* = 9), fibromyalgia (*n* = 8). (6) Other conditions (*n* = 17, 5%): organ transplants (*n* = 3), Cushing’s syndrome (*n* = 1): epilepsy (*n* = 1), eye conditions (*n* = 1), Alzheimer’s disease (*n* = 1), smoking (*n* = 1), non-cancerous prostate conditions (*n* = 2), not specified (*n* = 7).

About two-thirds (65%, *n* = 201) of the participants had been dealing with their medical condition for over five years, and the vast majority (86%, *n* = 267) received medical treatment for their condition. Frequency of medical appointments (before the outbreak) ranged from weekly (10.7%, *n* = 31) to yearly (8.6%, *n* = 25).

## 3. Results

Table 2 provides an answer to the first research question. About half of the participants reported a decline in mental and physical health during the first month of the COVID-19 outbreak. Regarding changes in physical SRH, 8% reported feeling greatly worsened, 39% reported feeling slightly worse, 44% reported feeling no change, 5% reported feeling slightly better, and 4% reported feeling greatly improved. Regarding changes in mental SRH, 10% reported feeling greatly worsened, 41% reported feeling slightly worse, 42% reported feeling no change, 4% reported feeling slightly better, and 3% reported feeling greatly improved. Regarding changes in general SRH, 7% reported feeling greatly worsened, 20% reported feeling slightly worse, 60% reported feeling no change, 8% reported feeling slightly better, and 5% reported feeling greatly improved.

Table 3 provides an answer to the second research question by demonstrating changes in health behaviors and loneliness. One month into the reinforcement of social distancing, participants reported a decrease in their perceived disease management abilities (T = −5.42, *p* < 0.001), and a decline in physical activity (T = 4.51, *p* < 0.001). In addition, total time spent on social media (T = 13.29, *p* < 0.001) and on online health communities (T = 3.76, *p* < 0.001) increased significantly. Over 50% of the participants reported eating more than they did before the outbreak. Furthermore, one month into the reinforcement of social distancing, participants reported a higher sense of loneliness (T = 12.76, *p* < 0.001). Whereas only one-third of the participants reported that they experienced loneliness before the outbreak, as many as two-thirds of them reported feeling lonely one month after the reinforcement of social distancing. Specifically, in regard to the question “How often do you feel that you lack companionship?” only 6% of the participants reported almost always before the outbreak, relative to as many as 27% during the outbreak. Similarly, in regard to the question “How often do you feel left out?” only 7% of the participants reported almost always before the outbreak, relative to as many as 20% during the outbreak. Finally, in regard to the question “How often do you feel isolated from others?” only 7% of the participants reported almost always before the outbreak, relative to as many as 21% during the outbreak. 

Table 4 provides an answer to the third research question by demonstrating differences in background variables and SRH measures between each of the five medical condition categories. There were no group differences in current physical health (F = 0.49, *p* = 0.79), but current mental SRH was lowest among participants with mental health conditions (F = 5.64, *p* < 0.001). Similarly, the feeling of loneliness was highest among participants with mental health conditions (F = 4.57, *p* < 0.001). 

Table 5 provides an answer to the fourth research question by presenting three ordinal logistic regressions to predict changes in SRH. The first model shows that a decline in physical SRH was predicted by crowded housing conditions (*p* = 0.005), higher BMI (*p* = 0.006), and higher frequency of medical visits before the crisis (*p* = 0.011). In addition, participants with orthopedic/pain conditions were more likely to experience a decline in their physical SRH relative to participants with mental (*p* = 0.002), metabolic (*p* = 0.008), and cardiovascular (*p* = 0.022) conditions. The second model shows that a decline in mental SRH was predicted by female gender (*p* = 0.049), crowded housing conditions (*p* = 0.002), and higher BMI (*p* = 0.005). In addition, participants with orthopedic/pain conditions were more likely to experience a decline in their mental SRH relative to participants with metabolic (*p* = 0.025) conditions. The third model shows that a decline in general SRH was predicted by female gender (*p* = 0.016), higher education (*p* = 0.015), crowded housing conditions (*p* = 0.001), and illness duration (*p* = 0.010).

## 4. Discussion

The present study investigated how people with chronic medical conditions perceived their health status one month into the reinforcement of social distancing due to the COVID-19 outbreak. We conducted an online survey among more than 300 people with chronic illnesses exactly one month (20–22 April 2020) after the Israeli government reinforced severe social-distancing regulations, and two days before these regulations were reduced. The survey included self-report information about SRH, health behaviors, and loneliness. Several interesting findings were identified. First, as many as about half of the participants with a chronic medical condition reported a decline in their physical SRH (47.2%) or mental SRH (50.5%) during the first month of the social distancing reinforcement. Second, a significant deterioration in health behaviors was reported, including a decrease in vegetable consumption and physical activity, an increase in time spent on social media, and a perception among about half of the participants (50.2%) that they were eating more than they had been before. Third, whereas only one-third of the participants reported that they had felt lonely before the outbreak, as many as two-thirds of the participants reported feeling lonely one month after the reinforcement of social distancing. Last, a decline in general SRH was linked with female gender, lack of higher education, crowded housing conditions, longer illness duration, and loneliness.

Data indicated that as many as about half of the participants in this study experienced a decline in their SRH after one month of social distancing reinforcement (first research question). To the best of our knowledge, the current study is the first to report changes in SRH during the coronavirus crisis among people with chronic illnesses [33,34,35,36]. Contrary to the current findings, in a study by Lei et al. (2020), conducted during the lockdown among 1593 healthy Chinese individuals, only 2.4% perceived their SRH as bad or very bad, and over two-thirds perceived their SRH as good or very good [46]. This difference may be explained by the fact that our study exclusively included participants with chronic medical conditions, who likely ordinarily have poorer SRH than do healthy individuals [47], whereas the Chinese sample was recruited from the general population. Furthermore, this discrepancy between our findings and those of Lei et al. (2020) may imply that the outbreak and the reinforcement of social distancing may have had a stronger negative impact on people with a chronic illness. Relatedly, Wang et al. (2020), who collected data at two timepoints during the COVID-19 outbreak [48], reported that poor SRH status, physical symptoms, and a history of chronic illness may have contributed to higher levels of stress, anxiety, and depression. It could be that there was a vicious cycle in which during the COVID-19 outbreak, chronic illness and poor SRH contributed to higher levels of stress, anxiety, and depression, which contributed (in a circular manner) to a decline in SRH.

Findings suggest a deterioration in health behaviors, including less exercising, less consumption of fresh fruits, and more time spent on social media (second research question). These findings can be explained by the limitations imposed by the social distancing reinforcement: People were not allowed to venture more than 100 m from their homes, and health-clubs, gyms, and recreational parks, as well as many stores and markets, were closed [3]. Although supermarkets and pharmacies did remain open, and local shortages of fresh produce were unusual [49], people with chronic medical conditions were discouraged from leaving the house [50], making it difficult to purchase fresh, healthy food. The reported decline in people’s ability to optimally manage their illnesses may have been related to the disruption in healthcare access, including the transition to online medical care and the absence of non-emergency medical surgeries due to the outbreak [51]. These disruptions may have posed specific difficulties (e.g., cancellations of routine medical appointments) for people with chronic medical conditions. Some people may also have had poor access to or difficulties in mastering the new technologies in remote care that were introduced by healthcare providers [52]. Several studies have already examined compliance to health behaviors directly related to COVID-19 (i.e., washing hands, avoiding social gatherings, self-isolating) [53,54], but limited information has been available regarding routine health behaviors such as nutrition, exercise, or pre-existing disease management during the pandemic among people with chronic illnesses. A single study identified health-related behaviors significantly associated with mental health among people in quarantine due to COVID-19 in Brazil: diet, tele-psychotherapy participation, and exercise level. The results from Brazil highlighted the role of health behaviors during the outbreak by showing that specific health behaviors, including balanced meals, exercising, and the use of tele-psychotherapy, may have impacted stress, depression, and anxiety levels [55]. 

Our findings demonstrated some similarities and some dissimilarities between people with different medical conditions in terms of SRH (third research question). For example, the adjusted model suggests that people with orthopedic/pain conditions experienced more deterioration in physical SRH relative to people with other medical conditions (i.e., mental, metabolic, or cardiovascular conditions). However, medical condition did not contribute to the variability we detected in general SRH declines. In Israel and the rest of the world, pain treatment and physical therapy centers closed their doors following the outbreak of the COVID-19 pandemic. Although telemedicine is being used to treat patients with many different medical conditions, pain management providers face a challenge in delivering services through video or other eHealth methods [51], preventing people with orthopedic/pain conditions from participating in treatments such as physical therapy. Moreover, the decline in physical activity reported in the entire sample may have more negatively affected people with orthopedic/pain conditions than those without, as these individuals are more vulnerable to lessened muscular strength, muscular endurance, and joint flexibility [56,57]. 

As expected, we found that participants felt lonelier during the first month of the outbreak than they had previously, and that loneliness was an important contributor to a decline in SRH (fourth research question). This finding is not surprising given that previous studies have already highlighted the importance of loneliness in health [18,20]. For example, a longitudinal study conducted before the coronavirus outbreak among a U.S. nationally representative sample reported that loneliness was associated with an increased mortality risk over a six-year period, and that this association (between loneliness and mortality) was explained by health outcomes [22]. Furthermore, a study conducted during the COVID-19 lockdown period in Spain found that higher reported loneliness was associated with higher distress [24]. In line with findings from the current study, Brodeur et al. (2020) used Google Trends to show a significant increase in searches for loneliness, worry, and sadness before and during the lockdown in Europe and the U.S. [58].

In the current study, the “crowded housing conditions” variable was a strong predictor of decline in both physical and mental SRH (fourth research question). Although a lower number of people living in the household is often correlated with worse SRH [26], even after adjusting for socioeconomic status, it could be that the social distancing regulations that forced people to stay in their homes for an extended period of time increased the negative effect of crowded housing conditions on participants’ health. In line with the current findings, Wang et al. (2020) reported that respondents staying in a household with three or more people one month into the COVID-19 outbreak had significantly higher posttraumatic stress disorder scores compared to respondents who lived alone [49]. Furthermore, female gender, higher BMI, higher frequency of medical visits (before the outbreak), and longer illness duration were also linked with a decline in SRH. These findings are not surprising given that females may be more vulnerable to psychological distress than are males [27], and this psychological distress may be linked with a stronger decline in mental SRH. Indeed, Mazza et al. [28] found that female gender was associated with higher levels of depression, anxiety, and stress during the COVID-19 lockdown in Italy [28]. Moreover, it has been reported that the COVID-19 pandemic has increased the care burden of women, negatively impacting women and their families [59]. In addition, higher BMI, more medical visits (before the outbreak), and longer illness duration are all likely to be linked with illness severity [60,61]. Therefore, our findings may imply that among people with a chronic illness, poorer medical condition (as indicated by higher BMI, more medical visits, and longer illness duration) is linked with a stronger decline in SRH. Future studies are warranted to clarify the long-term effects of social distancing on the health of this vulnerable population.

This study has several strengths, including the investigation of people with chronic illnesses exactly one month after the Israeli government decided to carry out an aggressive response to the COVID-19 outbreak. We used real-time data (gathered over three days) from a relatively large number of respondents at the peak of the COVID-19 epidemic and reinforcement of social distancing, thereby contributing preliminary information regarding the effect of such regulations on people suffering from chronic medical conditions. Several limitations should be noted. First, although participants were asked to report about their past and present situation, the cross-sectional design of the study does not allow us to determine causality. Second, participants were asked to provide retrospective data about their health status prior to the lockdown, as we did not have access to their pre-pandemic records. A recent study [62] examining the validity of a retrospective measurement (recall) of health, by using a test-retest design to measure reliability and agreement between prospective and retrospective patient-reported health, suggests that a retrospective measurement of health is a valid alternative to using reference data for the purpose of estimating past health status. Third, we relied on self-reported data and not on clinical records. For example, height and weight were assessed by self-report, which is likely less accurate than objective measurement. However, studies show that web responders usually provide accurate information about themselves [63]. Relatedly, we do not have information about participants’ specific medical diagnoses, and it may be that such information could have provided important insights into the detected variability in SRH. In addition, SRH is a subjective measure and may not always reflect objective measures of health, although a strong predictive capacity regarding various health-related outcomes is well established [6,7,8]. Fourth, the sampling design is subject to biases of internet-based surveys. Participants voluntarily selected to take the survey online, and although the current sample was large and heterogeneous, it was not a representative sample.

## 5. Conclusions

Findings highlight the important role of loneliness in health [58], while demonstrating how the near-lockdown measures, announced by the Israeli government in response to the COVID-19 outbreak, may have had a negative impact on the health of people with chronic illnesses. Future longitudinal studies are warranted to clarify the long-term effects of loneliness on the health of people with chronic medical condition during the COVID-19 outbreak.

## Figures and Tables

**Table 1 ijerph-17-05403-t001:** Demographic and medical characteristics.

Characteristics		
Gender	% (n), Female	59.5 (178)
	% (n), Male	40.5 (121)
Age ^2^	% (n), 18–45	19.2 (60)
	% (n), 46–55	13.8 (43)
	% (n), 56–65	22.1 (69)
	% (n), 66–75	34.3 (107)
	% (n), ≥76	10.6 (33)
Marital status ^2^	% (n), Married	37.5 (118)
	% (n), Unmarried	62.5 (197)
Education	% (n), 12 years or less	26.8 (84)
	% (n), Vocational education	24.6 (77)
	% (n), University degree	48.6 (152)
Work status (before COVID-19) ^2^	% (n), Employed	42.8 (131)
	% (n), Unemployed	57.2 (175)
Work status (during COVID-19) ^2^	% (n), Employed	29.9 (83)
	% (n), Unemployed	70.1 (195)
Economic status ^2^	% (n), Bad/very bad	28.8 (90)
	% (n), pretty good	35.3 (110)
	% (n), Good/very good	35.9 (112)
Religiosity	% (n), Secular	70.9 (222)
	% (n), Traditional	18.8 (59)
	% (n), Religious	10.2 (32)
Num. of people in the household ^2^	% (n), One	18.5 (58)
	% (n), Two	47.0 (147)
	% (n), More than two	34.5 (108)
Crowded housing conditions ^1^	Average (Std)	1.50 (1.05)
Main medical condition	% (n), Mental health	11.7 (36)
	% (n), Metabolic	27.0 (83)
	% (n), Cardiovascular	17.3 (53)
	% (n), Cancer & autoimmune	20.8 (64)
	% (n), Orthopedic/pain	17.6 (54)
	% (n), Other	5.5 (17)
Duration of medical condition	% (n), <1 year	7.2 (23)
	% (n), 1–3 years	16.7 (52)
	% (n), 3–5 years	11.3 (35)
	% (n), >5 years	64.6 (201)
Receive medical care for the condition	% (n), Yes	85.6 (267)
	% (n), No	14.4 (45)
Medical visit frequency ^2^	% (n), Weekly or more	10.7 (31)
	% (n), Monthly	33.8 (98)
	% (n), Every six months	46.2 (134)
	% (n), Yearly or less	8.6 (25)
BMI	Average (Std)	27.4 (6.2)

BMI = Body Mass Index. ^1^ Items rated on a 5-point scale; higher scores represent a more crowded living perception. ^2^ Original categories combined due to few responses.

**Table 2 ijerph-17-05403-t002:** Self-rated health (SRH) measures.

SRH Measures	Title	% (*n*)
Current Physical SRH	Bad/very bad	14.6 (46)
	Neither good nor bad	46.7 (147)
	Good/very good	38.7 (122)
Current Mental SRH	Bad/very bad	14.2 (44)
	Neither good nor bad	30.3 (94)
	Good/very good	55.5 (172)
Change in Physical SRH during the outbreak	Improved	8.9 (28)
	Unchanged	43.9 (138)
	Worsened	47.2 (148)
Change in Mental SRH during the outbreak	Improved	7.3 (21)
	Unchanged	42.3 (129)
	Worsened	50.5 (154)
Change in general SRH during the outbreak	Improved	12.5 (39)
	Unchanged	60.1 (187)
	Worsened	27.4 (85)

SRH = Self-Rated Health.

**Table 3 ijerph-17-05403-t003:** Changes in health behaviors and loneliness one month into the COVID-19 outbreak among 315 adults with chronic medical conditions.

	Range	Before (Average, Std)	During (Average, Std)	T-Test	*p*-Value
Disease management ^1^	1–5	3.9 (1.0)	3.6 (1.1)	5.42	<0.001
Time spent on social media (hours/day) ^2^	1–7	3.2 (1.1)	3.9 (1.2)	13.29	<0.001
Time spent on online health communities (hours/day) ^2^	1–7	2.0 (0.7)	2.2 (0.9)	3.76	<0.001
Physical activity (times/week) ^3^	0–10	3.5 (2.4)	2.8 (2.4)	4.51	<0.001
Fruit consumption (units/day)	0-10	2.7 (2.0)	2.6 (2.0)	1.89	0.060
Vegetable consumption (units/day)	0–10	3.2 (2.4)	3.0 (2.3)	2.66	0.008
Loneliness (Total score) ^4^	3–9	4.3 (1.7)	5.6 (2.0)	12.76	0.001
Food Consumption ^5^	% (*n*) Much more than before			19.7 (62)	
	% (*n*) A little more than before			30.5 (96)	
	% (*n*) Same as before			40.0 (126)	
	% (*n*) A little less than before			7.0 (22)	
	% (*n*) Much less than before			2.9 (9)	

^1^ “In my opinion, I manage my illness optimally” with answers ranging from 1 (strongly disagree) to 5 (strongly agree). ^2^ Answers ranging from 1 (not at all) to 7 (more than 10 hours per day) ^3^ Half an hour or more of physical activity. ^4^ Revised UCLA Loneliness Scale. ^5^ “Relative to the amount of food I consumed before the outbreak, I estimate that I currently consume” 1 (a lot more than before) to 5 (a lot less than before).

**Table 4 ijerph-17-05403-t004:** Background, Loneliness, and Self-Rated Health (SRH) by Medical Condition Category.

Background, Loneliness, and Self-Rated Health	Mental *n* = 36	Metabolic *n* = 83	Cardiovascular *n* = 53	Autoimmune *n* = 64	Orthopedic/Pain *n* = 54	Test-Statistics
**Background Characteristics**						
Age (years), average (Std)	42.3 (14.3) ^a^	63.1 (13.1) ^c^	65.1 (16.1) ^c^	55.2 (16.0) ^b^	57.8 (14.2) ^b^	16.35, *p* < 0.001
Female, % (n)	71.43 (25) ^c^	43.04 (34) ^b^	30.43 (14) ^a^	79.03 (49) ^c^	70.37 (38) ^c^	39.40, *p* < 0.001
BMI, average (Std)	25.58 (6.26) ^a^	30.30 (7.38) ^b^	27.58 (4.48) ^a^	25.87 (4.52) ^a^	25.99 (5.10) ^a^	5.61, *p* < 0.001
Duration, average (Std)	5.46 (1.15) ^ab^	5.49 (0.92) ^b^	5.09 (1.01) ^a^	5.22 (1.06) ^ab^	5.07 (1.42) ^a^	1.54, *p* = 0.179
**Current SRH and Loneliness**, average (Std)						
SRH physical (current) ^1^	3.08 (0.97)	3.29 (0.82)	3.28 (0.86)	3.25 (0.67)	3.17 (0.86)	0.49, *p* = 0.787
SRH mental (current) ^1^	2.92 (1.03) ^a^	3.63 (0.96) ^b^	3.75 (0.90) ^b^	3.66 (0.85) ^b^	3.26 (0.81) ^b^	5.64, *p* < 0.001
Loneliness (current) ^2^	6.75 (2.11) ^c^	5.46 (1.81) ^ab^	5.24 (2.09) ^a^	5.66 (2.15) ^ab^	6.00 (1.90) ^b^	4.57, *p* < 0.001
**Changes in SRH**, average (Std)						
Change in physical SRH ^3^	3.36 (0.96) ^ab^	3.29 (0.93) ^a^	3.34 (0.81) ^a^	3.53 (0.78) ^ab^	3.70 (0.82) ^b^	1.91, *p* = 0.092
Change in mental SRH ^3^	3.72 (1.03) ^b^	3.35 (0.91) ^a^	3.48 (0.70) ^ab^	3.48 (0.77) ^ab^	3.73 (0.64) ^b^	1.96, *p* = 0.084
Change in general SRH ^3^	3.22 (1.02) ^ab^	3.13 (0.79) ^ab^	3.19 (0.66) ^ab^	3.08 (0.82) ^a^	3.40 (1.01) ^b^	0.96, *p* = 0.440

SRH = Self-Rated Health. ^abc^ Different superscript values indicate that the absolute values differ significantly. ^1^ Items rated on a scale from 1 (very bad) to 5 (very good). ^2^ Total score; items rated on a scale from 1 (hardly ever) to 3 (often). ^3^ Items rated on a scale from 1 (greatly improved) to 5 (greatly worsened).

**Table 5 ijerph-17-05403-t005:** Ordinal logistic regression predicting decline in self-rated health (SRH) one month into the Covid-19 outbreak among 292 people with chronic illnesses.

Independent Variables		Decline in Physical SRH ^1^				Decline in Mental SRH ^2^				Decline in General SRH ^3^			
		EXP (B)	*p*-Value	95% Wald Confidence Interval for Exp(B)		EXP (B)	*p*-Value	95% Wald Confidence Interval for Exp (B)		EXP (B)	*p*-Value	95% Wald Confidence Interval for Exp (B)	
				Lower	Upper			Lower	Upper			Lower	Upper
Female gender		1.04	0.893	0.60	1.79	**1.79**	**0.049**	**1.00**	**3.19**	**2.07**	**0.016**	**1.15**	**3.74**
Age		1.04	0.734	0.84	1.29	0.93	0.524	0.74	1.16	1.07	0.564	0.86	1.33
Higher education		1.02	0.938	0.57	1.85	0.85	0.610	0.45	1.59	**2.20**	**0.015**	**1.16**	**4.18**
Crowded housing conditions		**0.39**	**0.005**	**0.21**	**0.76**	**0.35**	**0.002**	**0.17**	**0.69**	**0.32**	**0.001**	**0.16**	**0.64**
BMI		**1.07**	**0.006**	**1.02**	**1.12**	**1.07**	**0.005**	**1.02**	**1.12**	1.04	0.107	0.99	1.09
Medical visit frequency ^4^		**0.68**	**0.011**	**0.51**	**0.92**	0.84	0.258	0.62	1.14	0.83	0.255	0.60	1.14
Illness duration		0.93	0.577	0.74	1.19	1.13	0.353	0.87	1.47	**1.43**	**0.010**	**1.09**	**1.88**
Medical condition ^5^	Mental	**0.19**	**0.002**	**0.07**	**0.55**	0.53	0.248	0.18	1.55	0.40	0.105	0.13	1.21
	Metabolic	**0.34**	**0.008**	**0.16**	**0.76**	**0.40**	**0.025**	**0.18**	**0.89**	0.77	0.527	0.33	1.76
	Cardiovascular	**0.36**	**0.022**	**0.15**	**0.86**	0.71	0.457	0.29	1.75	1.27	0.619	0.49	3.28
	Autoimmune	0.62	0.235	0.29	1.36	0.47	0.066	0.21	1.05	0.66	0.329	0.28	1.53
	Orthopedic/pain	Ref.				Ref.				Ref.			
Loneliness—current		**1.23**	**0.002**	**1.08**	**1.40**	**1.58**	**0.000**	**1.37**	**1.82**	**1.21**	**0.008**	**1.05**	**1.39**

SRH = Self-Rated Health. BMI = Body Mass Index. Significant values appear in bold. ^1^ “How much do you feel that the COVID-19 crisis has affected your physical health?” with answers ranging from 1 (greatly improved) to 5 (greatly worsened).^2^ “How much do you feel that the COVID-19 crisis has affected your mental health?” with answers ranging from 1 (greatly improved) to 5 (greatly worsened). ^3^ “Has your medical condition improved or worsened in the last month?” with answers ranging from 1 (greatly improved) to 5 (greatly worsened). ^4^ Before the outbreak. ^5^ Participants with “other” medical condition were omitted from the analyses.

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
