# Peer review of "One Month into the Reinforcement of Social Distancing due to the COVID-19 Outbreak: Subjective Health, Health Behaviors, and Loneliness among People with Chronic Medical Conditions"

_ijerph, 2020, doi:10.3390/ijerph17155403_

Round 1
Reviewer 1 Report
Important and interesting study. Some points:
Procedure: Please make more clear that this is a convenience sample (also in the abstract)
Please include more fetails regarding the ethics statement
Procedure: Any inclusion or exclusion criteria?
Medical condition: Is this a validated instrument?
Health behaviors: Is this a validated instrument? (same question for disease management; social media consumption)
Retrospective assessment of health. This is a main limitation of this study. Please markedly extend the limitations section.
Did you really assess loneliness before the COVID-19 crisis (page 4, line 156)? This is a bit confusing (since this is a cross-sectional survey). Retrospective?
Results/Discussion: Well done.
Limitations are mainly adequately addressed. However, please also refer to the retrospective assessment (likely to be biased)
Conclusions: Appropriate.
Tables: Well done.
Reviewer 2 Report
This manuscript describes a cross-sectional survey collected through virtual health communities in Israel one month after the government instituted lock-down measures related to the novel coronavirus. The survey for current and past perceived ratings of health, health-related behaviors and loneliness was computed by 315 respondents with a variety of chronic diseases.
While little is known about the relationship among these factors in this population during this unprecedented time, and the information presented is helpful in understanding potential outcomes of these governmental actions on other aspects of health, the study design is weak, especially in asking for retrospective subjective data. The authors note some of these weaknesses as limitations and that additional longitudinal studies should be carried out although the feasibility of this is in doubt.
The introductions provides 4 research questions that were to be addressed by the survey. There is insufficient foundation provided as to why these questions were being asked. Additional background related to the role of loneliness in chronic health conditions. Line 312 in discussion notes, “as expected…” suggests a rationale, it should be introduced in the background. Further, it is unclear that some of the health behaviors would impact the variety of chronic health conditions equally. Further, as two of the health behaviors questions related to consumption of fresh fruits and vegetables, the authors should address if there were changes in availability of these foods during the lock-down. And, why the limit of 10 servings per week (possible ceiling response bias).
Please clarify if participants self-selected their main medical condition or if they were screened for single health conditions. It seems likely that many of these conditions could be present together.
Please structure the methods, results and discussion to align with the research questions more closely. Table 5 appears to address a hypothesis rather than a research question (#3). Eliminate what is beyond the scope of these questions unless rationale for their inclusion is provided. If demographics are included in analyses, a rationale for their inclusion should be provided.
Lines 336-354of discussion– consider the greater household/child care burdens that fell more on women than men during this time as has been reported.
It was helpful to remind the reader of data source (question) in table footnotes.
Minor:
Please list response options in the order they were presented to the participants: eg, bad, very bad, quite good, good, very good (line 115 and Table 1).
Helpful to indicate what the number presented in parentheses in tables are – Std dev usually.
The manuscript could be shortened by not duplicating some of the information in the tables in the text. Rather just summarize in the text.
Reviewer 3 Report
The manuscript describes a cross-sectional research study aimed to investigate whether people with chronic medical conditions experienced changes in Self-Rated Health (SRH), in health behaviors and loneliness, after one month of social distancing reinforcement announced by the Israeli government due to the COVID-19 outbreak. In general, I find this article to be well written, I do find this paper to be a good discussion starter and would support publishing this manuscript with minor revision.
I would ask the Authors to address minor amendments, as follows:
- Improve Introduction section
- Minor text editing

Reviewer 4 Report
Thank you for your interesting work. The champion is limited, as you know, so this could be the first of other inquiries during this period.
Be aware of some little corrections: line 47-48 references
line 73 - add further references on this topic there are recent publications/reports
Round 2
Reviewer 2 Report
This manuscript is much improved and is easier to follow with the edits by the authors. One limitation is that respondents were asked to self-select their major health condition. As many chronic conditions are concurrently present, and others have reported on the impact of multiple chronic diseases on mental health and quality of life, it is important that this be noted as a limitation in the discussion section.